# Structural basis of sodium ion-dependent carnitine transport by OCTN2

James S. Davies [1,7], Yi C. Zeng [1,2,7] ✉, Chelsea Briot[3], Simon H. J. Brown [4,5], Renae M. Ryan [3,6] ✉ & Alastair G. Stewart [1,2] ✉

Carnitine is essential for the import of long-chain fatty acids into mitochondria, where they are used for energy production. The carnitine transporter OCTN2 (novel organic cation transporter 2, SLC22A5) mediates carnitine uptake across the plasma membrane and as such facilitates fatty acid metabolism in most tissues. OCTN2 dysfunction causes systemic primary carnitine deficiency (SPCD), a potentially lethal disorder. Despite its importance in metabolism, the mechanism of high-affinity, sodium ion-dependent transport by OCTN2 is unclear. Here we report cryo-EM structures of human OCTN2 in three conformations: inward-facing ligand-free, occluded carnitine- and Na⁺-bound, and inward-facing ipratropium-bound. These structures define key interactions responsible for carnitine transport and identify an allosterically coupled Na⁺ binding site housed within an aqueous cavity, separate from the carnitine-binding site. Combined with electrophysiology data, we provide a framework for understanding variants associated with SPCD and insight into how OCTN2 functions as the primary human carnitine transporter.

ʟ-carnitine is an essential molecule that facilitates the transport of fatty acids into mitochondria for β-oxidation (Fig. 1a)[1]. In humans, the primary source of carnitine is the diet, although it can also be endogenously synthesised and distributed throughout the body[2]. OCTN2 (novel organic cation 2, SLC22A5), is the primary transporter responsible for high-affinity uptake of carnitine and is highly expressed in tissues with high rates of fatty acid metabolism such as heart, skeletal muscle and brain[3,4]. OCTN2 is also critical for efficient renal tubular reabsorption of carnitine, where the vast majority of filtered carnitine (and short-chain acyl derivatives of carnitine) are reabsorbed[5–7]. Mutations in the gene encoding OCTN2, *SLC22A5*, can impair fatty acid metabolism and lead to systemic primary carnitine deficiency (SPCD, OMIM 212140)[8], an autosomal recessive disorder whose clinical manifestations include cardiomyopathy, hypoglycemia, chronic muscle weakness and liver dysfunction[9]. Furthermore, single nucleotide polymorphisms in OCTN2 are associated with increased susceptibility to Crohn's inflammatory bowel disease[10]. Although recent comprehensive functional genomics work has shed light on how OCTN2 missense variants affect both carnitine transport and membrane localisation[11], the mechanism of carnitine transport and how clinically relevant variants affect transport at a molecular level are yet to be determined.

OCTN2 is a member of the SLC22 class of the solute carrier (SLC) transporter family that encompasses organic cation transporters (OCTs) and organic anion transporters (OATs). These transporters all adopt the major facilitator superfamily (MFS) fold and are capable of translocating a range of different molecules across the plasma membrane, including endogenous substrates, clinically relevant drugs and xenobiotics[12]. The OCTN subclade, comprised of OCTN1 (SLC22A4) and OCTN2, are a distinct subgroup of OCTs that

[1]Molecular, Structural and Computational Biology Division, The Victor Chang Cardiac Research Institute, Darlinghurst, NSW, Australia. [2]School of Clinical Medicine, Faculty of Medicine and Health, UNSW Sydney, Sydney, NSW, Australia. [3]School of Medical Sciences, Faculty of Medicine and Health, University of Sydney, Sydney, NSW, Australia. [4]ARC Industrial Transformation Training Centre for Cryo-Electron Microscopy of Membrane Proteins, University of Wollongong, Wollongong, NSW, Australia. [5]Molecular Horizons and School of Science, University of Wollongong, Wollongong, NSW, Australia. [6]School of Biomedical Engineering, Faculty of Engineering, University of Sydney, Sydney, NSW, Australia. [7]These authors contributed equally: James S. Davies, Yi C. Zeng. ✉e-mail: y.zeng@victorchang.edu.au; renae.ryan@sydney.edu.au; a.stewart@victorchang.edu.au

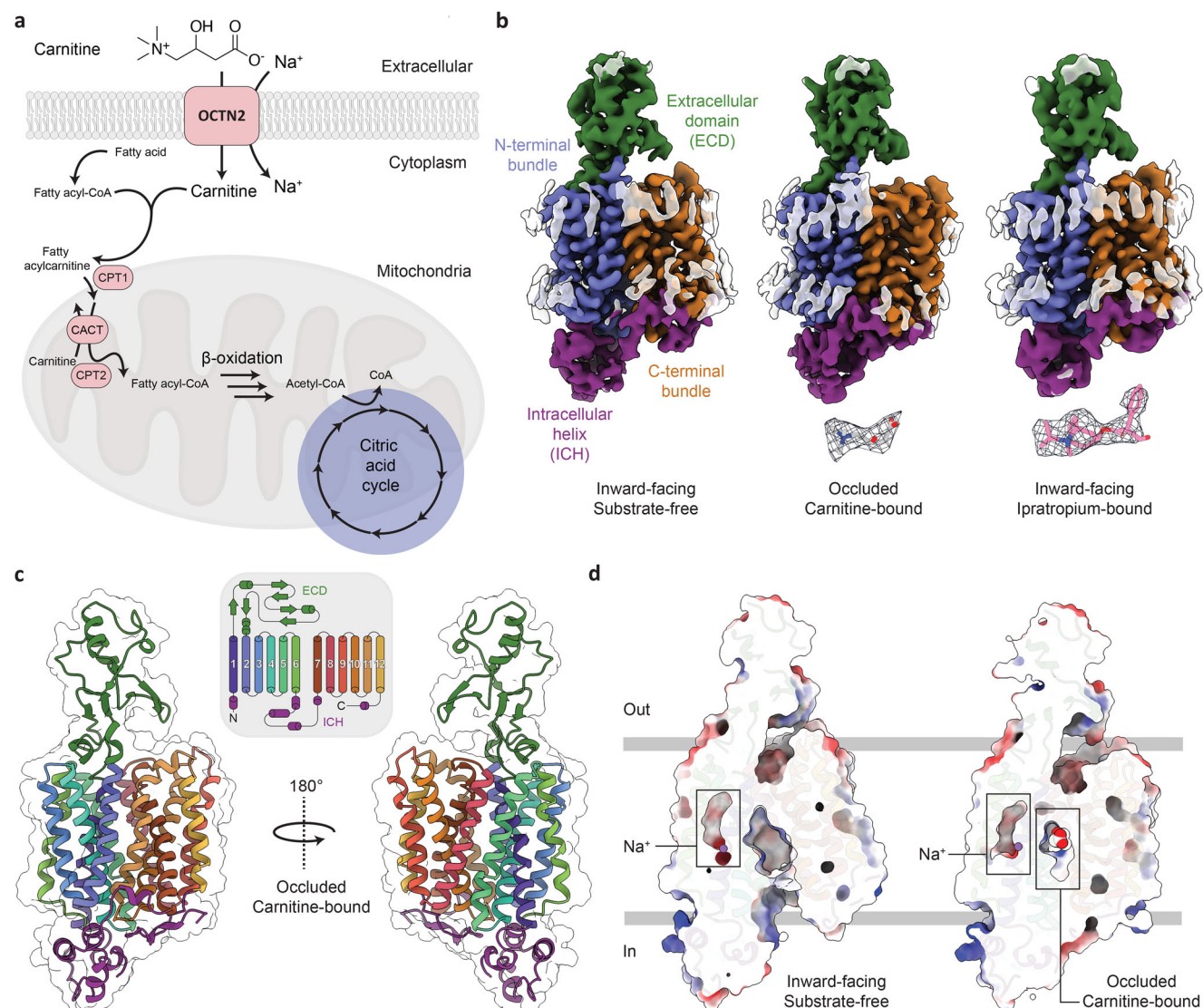

**Fig. 1 | Structures of OCTN2 in the inward-facing and occluded conformations. a** OCTN2 facilitates the Na⁺-dependent transport of carnitine into the cell, which is then used for the transport of fatty acids (via the carnitine palmitoyltransferase I-containing complex; CPTI, and the carnitine-acylcarnitine translocase; CACT) to the mitochondrial matrix for β-oxidation. **b** Cryo-EM maps of OCTN2 in the inward-facing substrate-free (*left*), occluded (*middle*) and inward-facing ipratropium-bound (*right*) conformations, contoured at 7.5σ as calculated by *ChimeraX*[61] and coloured according to domain: the intracellular helix; purple, the N-terminal transmembrane bundle; blue, the extracellular domain; green and the C-terminal transmembrane bundle orange. *Inset* Cryo-EM density of carnitine and ipratropium, from sharpened maps contoured at 8σ. **c** *Left:* Cartoon representations of the occluded conformation of OCTN2, with helices coloured as per the schematic above showing the topology. **d** Surface cutaways of the inward-facing (*left*) and occluded (*right*) conformations. Na⁺ in each conformation is shown in purple, within an enclosed cavity, as indicated. In the occluded conformation, carnitine is also bound within an enclosed cavity, which is separate from the Na⁺-cavity.

primarily catalyse the uptake of the zwitterionic metabolites ergothioneine[13] and carnitine respectively, but are also capable of transporting cations such as tetraethylammonium (TEA)[14,15]—indicating a potential dual transport function[15,16]. However, OCTN1 and OCTN2 both have narrower substrate specificity compared to polyspecific OCTs and OATs, and uptake assays performed with a large panel of drugs suggest that OCTN2 is not a general drug transporter[3,12,17,18]. Compounds that bear a quaternary or tertiary amine group (e.g. the bronchodilator ipratropium and the calcium channel blocker verapamil) have been proposed as substrates[19,20], though these compounds, alongside a number of other cationic drugs, were shown to be poor substrates for OCTN2[21]. Nevertheless, it is clear that OCTN2 can interact with a plethora of different compounds, many of which are able to inhibit carnitine uptake[22]— an important consideration given the broad tissue distribution of OCTN2 and its role in cellular homeostasis.

OCTN1 and OCTN2 are further differentiated from other SLC22 family members in that the transport of zwitterionic substrates is sodium ion (Na⁺)-dependent[3,6,13,23,24]. To date, conventional Na⁺ or H⁺ ion coupling has yet to be observed in OCTs or OATs, with OCTs thought to function as uniporters rather than symporters, and the OATs coupling the uphill transport of organic anions to the downhill movement of intracellular organic anions, such as α-ketoglutarate[25]. In vitro cellular and vesicular uptake assays[3,6,23] support the hypothesis that OCTN2 is a high-affinity secondary active transporter that utilises Na⁺ gradients to drive the uphill transport of carnitine into the cell. Typically, the concentration of carnitine in plasma is in the low μM range (~40–60), while tissue concentrations (taken here to approximate cytoplasmic concentrations) are in the low mM range (~1–5 mM)[26,27].

Here we report cryo-electron microscopy (cryo-EM) structures of human OCTN2 in a substrate-free inward-facing conformation, a

substrate-bound occluded conformation, and an ipratropium-bound inward-facing conformation, revealing how OCTN2 binds and transports carnitine. Both the occluded and the inward-facing structures show Na$^+$ coordinated deep within an aqueous cavity that is allosteric to the carnitine-binding site. Residues involved in carnitine and Na$^+$ coordination were evaluated using two electrode voltage-clamp electrophysiology, confirming Na$^+$-dependent carnitine transport and the functional significance of these sites. Comparison of these structures provides the molecular basis for carnitine transport by OCTN2 and a framework to better interpret variants associated with SPCD.

## Results

### Cryo-EM structure determination of OCTN2

To determine the structure of OCTN2, we overexpressed full-length human OCTN2 in HEK293F cells and isolated the transporter using a GFP-nanobody purification strategy[28]. The protein was solubilised using n-dodecyl β-D-maltoside (DDM) and cholesteryl hemi-succinate (CHS), which was then exchanged for glyco-diosgenin (GDN) during purification. Size-exclusion chromatography yielded monodisperse protein that was suitable for cryo-EM grid preparation and screening (Supplementary Fig. 1). We solved cryo-EM structures of OCTN2 (-62.7 kDa) in a substrate-free inward-facing conformation to 3.0 Å resolution, a carnitine-bound occluded conformation to 2.7 Å resolution (Fig. 1b, Supplementary Table 1) and an ipratropium-bound inward-facing conformation to 3.1 Å resolution (Supplementary Figs. 1 and 2).

In the absence of carnitine, the structure of OCTN2 was observed in an inward-facing conformation (Fig. 1b), with a large solvent accessible cavity (-1200 Å$^3$) located between the N- and C-terminal bundles (Fig. 1d). We describe the inward-facing structure as substrate-free rather than apo, as we observe weak, non-proteinaceous density in the cleft between the two domains (Supplementary Fig. 3). This density does not resemble carnitine and is consistent with weak densities observed in substrate-free OCT1 structures[29,30]. We hypothesise here that this density may be a detergent, a lipid molecule, or reflect an averaged density of multiple chemical species. OCTN2 has the archetypal MFS fold, with 12 transmembrane helices (TM) organised into two pseudo-symmetric 6-TM bundles (Fig. 1c), and an intracellular helical domain (ICH) that links TM6 of the N-terminal bundle and TM7 of the C-terminal bundle. The extracellular domain (ECD) is stabilised by two disulfide bonds between C50-C113 and C81–C136, as in human OCT1[29] and rat OAT1[25,31]; however, OCTN2 does not exhibit the distal disulfide seen in OCTs (C60-C121 in OCT1). The structures confirm that OCTN2 has three N-linked glycosylation sites situated on the ECD, with relatively strong density corresponding to glycans extending from N57, N64 and N91 (Supplementary Fig. 4). Although at lower thresholds, the glycan chain density extends out significantly (-20 Å) (Supplementary Fig. 4), because of the limited resolution, we have only modelled one N-linked GlcNAc unit at each position. The ECD, like in other SLC22 members, is thought to be linked with biogenesis and maturation of the protein to the plasma membrane. Glycosylation of the ECD does not explicitly affect maturation, though mutation of these three asparagine residues prevents trafficking to the plasma membrane[32]. Furthermore, a splice variant of OCTN2 that results in an insertion in the ECD leads to retention of the transporter in the endoplasmic reticulum[33]. Indeed, the structure of the ECD may affect transport itself (mutations in the ECD increase the $K_M$ for carnitine[32]), however, the structural basis for this is unclear.

### The carnitine bound OCTN2-occluded structure defines the substrate and sodium ion binding sites

In the presence of 10 mM L-carnitine the structure of OCTN2 was determined in the fully occluded, carnitine- and Na$^+$-bound conformation (Fig. 1). We observe strong density for carnitine bound within a central cavity situated between the N- and C-terminal

domains. The cavity has a volume of -370 Å$^3$, notably larger than that of carnitine, which we estimate at -150 Å$^3$. The cavity is fully closed to the solvent environment from both sides of the membrane. The structure clearly shows how OCTN2 coordinates both the negative and positively charged moieties of carnitine (Fig. 2a).

The carboxylate group of carnitine forms a salt bridge with the guanidinium group of R471 (TM11), as well as a hydrogen bonding interaction with the hydroxyl of Y447 (TM10), both residues from the C-terminal bundle. Additionally, a water molecule interacts with R471 and Q207 (TM4), connecting the carboxylate to the N-terminal bundle. The density for this water was weaker than that of the ligand, although strong enough to warrant modelling (Supplementary Fig. 3) with the local resolution range for this region of the map reaching -2.4 Å (Supplementary Fig. 1). Altogether, the carnitine carboxylate is fully engaged with stabilising interactions, which is consistent with the high-affinity nature of transport[3].

The positively charged ammonium group of carnitine is coordinated by conserved hydrophobic residues that form an aromatic cage around carnitine (Fig. 2a). Residues Y239 (TM5), Y358 (TM7) and F443 (TM10) all positioned ideally to form strong cation-$\pi$ interactions, with nearby residues Y211, F359, Y387 and Y447 positioned within reasonable distance to form weak cation-$\pi$ interactions. The hydroxyl groups of Y239 and Y358 are within hydrogen bonding distance to one another and form a part of the extracellular gate, enclosing the ammonium group from the extracellular side of the pocket. Furthermore, Y358 is locked in place by E383 and K436, an electrostatic pair conserved in OCTs and OATs[34] that is involved in the coordination and release of substrates within the pocket[29] (Supplementary Fig. 5). The opposite gating residues on the intracellular side are Y211 and Y447 which pack together to form a part of the intracellular gate. Near to this gate, we modelled two additional water molecules within the substrate-binding cavity, proximal to the ammonium side of carnitine, bridging between Y211 (TM4) and S231/C236 (TM5) (Supplementary Fig. 5). Overall, the occluded structure illustrates how OCTN2 accommodates the opposing charged moieties of carnitine, as well as an apparent secondary role for ordered solvent within the substrate-binding site.

In our carnitine-bound structure, we identify a Na$^+$ binding site sandwiched between TM1 and TM4 of the N-terminal bundle, which is further housed within an aqueous cavity that extends upward within the N-terminal domain (Figs. 1d and 2b). The *CheckMyMetal*[35] server supports our assignment of Na$^+$ and predictions made using *Alphafold3*[36] situate Na$^+$ in this position with high confidence (Supplementary Fig. 6). The Na$^+$ site is located -10 Å away from the substrate cavity, towards the bottom cytoplasmic side of a separate, enclosed aqueous cavity with a volume of -195 Å$^3$. The strong density is coordinated by the backbone carbonyl of S28, the sidechain oxygens of N32 (TM1) and N210 (TM4), as well as a water molecule in an axial position (Fig. 2a & Supplementary Fig. 3). Accordingly, electronegative potential concentrates at the base of the cavity where Na$^+$ is positioned (Fig. 2b) and the Na$^+$ ion is coordinated by four oxygen ligands. In addition, above the Na$^+$-binding site (in the cavity, toward the extracellular side) we observe a second density that we have modelled as a water molecule (Fig. 2b, labelled "W2" & Supplementary Fig. 3). The Na$^+$-OH$_2$ distance is -3.5 Å, suggesting it is engaged in a weak electrostatic interaction with the ion, that may help to further stabilise the Na$^+$-binding site. The density corresponding to the Na$^+$ is stronger compared to the observed water molecules throughout the Na$^+$-binding cavity, with the latter only becoming evident in sharpened maps (Supplementary Fig. 3). Unlike other Na$^+$-dependent MFS transporters, such as MelB[37] and MFSD2A[38–40], where the Na$^+$ site forms part of the substrate-binding cavity, the OCTN2 Na$^+$-binding site is buried within a solvent-filled cavity in the N-terminal bundle.

The Na$^+$-ion binding site is also present in the ligand-free inward-facing structure, with the same occluded binding cavity described

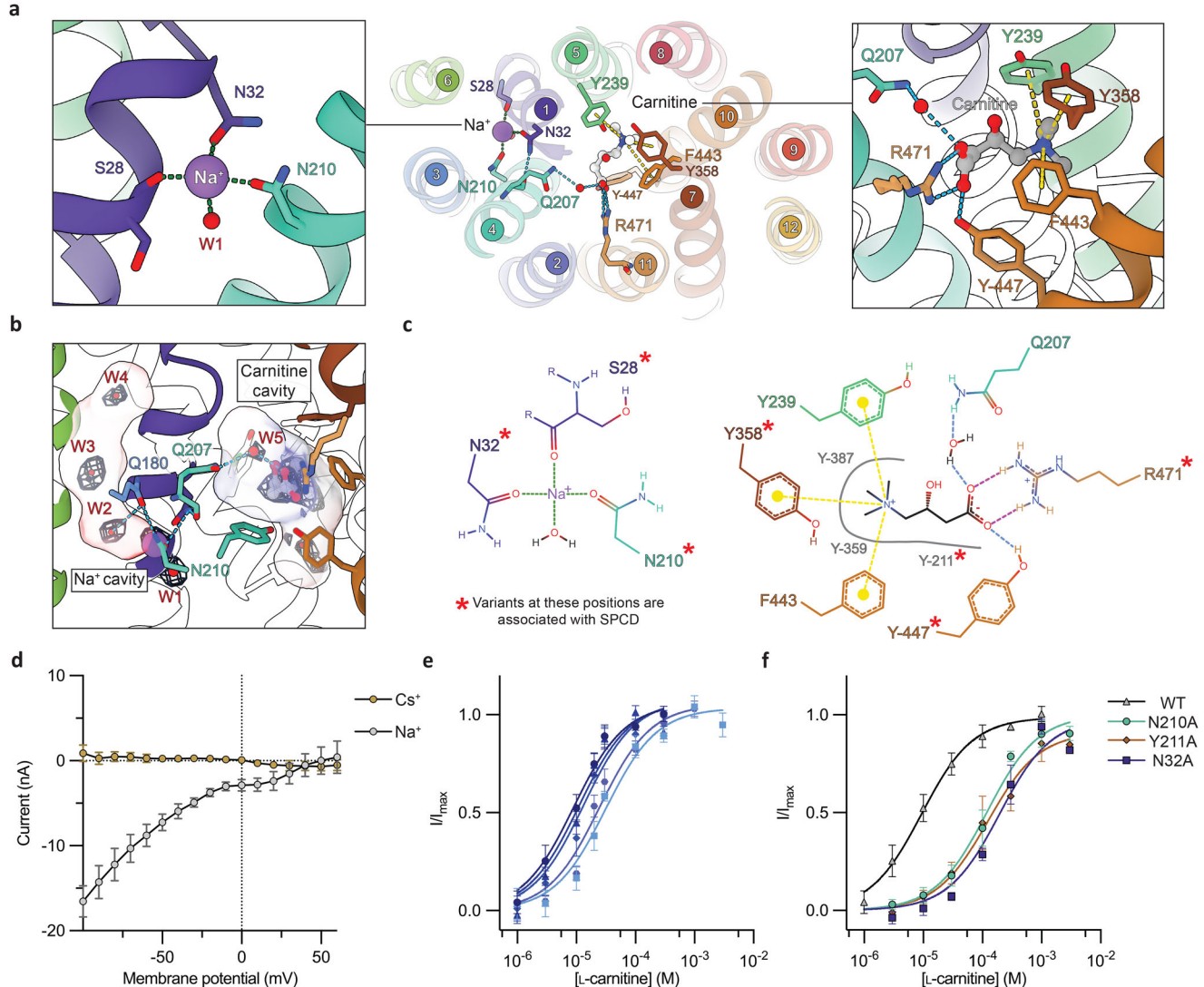

**Fig. 2 | The occluded conformation substrate- and Na⁺ ion-binding sites. a** View of the Na⁺- and carnitine- binding sites. *Left* Na⁺ coordinated by S28, N32 and N210, as well as a water molecule below (W1) *Middle* Cutaway view of OCTN2 looking down from the ECD, showing the central carnitine-binding site and the Na⁺-binding site situated in the N-terminal bundle. *Right* View of the carnitine-binding site, showing the ionic interaction between the carboxylate of carnitine and R471, hydrogen bonding interactions (blue dashes) with Y447 and Q207 via a water molecule, and the conserved aromatic cage around the positively charged quaternary ammonium moiety (cation- interactions shown by yellow dashes). Helices are coloured according to Fig. 1c. **b** Representation of the Na⁺ and carnitine binding cavities, coloured by electrostatic surface potential as calculated by *ChimeraX*, with cryo-EM density corresponding to carnitine, Na⁺ and water molecules (W2, W3, W4) shown as mesh contoured at 7.5σ. Residues binding Na⁺ and putative linking residues (Q207 and Y211) are depicted as sticks. **c** 2d representation of the Na⁺ and carnitine binding sites. Red asterisks indicate positions within each binding site

where variants are associated with SPCD per Koleske et al.[11]. **d** Current voltage (IV) relationships measured in oocytes expressing hOCTN2 upon application of 100 μM L-carnitine in the presence of either 100 mM Na⁺ (grey) or Cs⁺(gold). **e** Carnitine concentration-response relationships were measured by applying L-carnitine (1–6000 μM) to oocytes expressing hOCTN2 in the presence of differing Na⁺ concentrations (10, 30, 40, 60 and 100 mM), with darker blue curves corresponding to higher Na⁺. Currents elicited at each carnitine concentration were measured at −60 mV and fitted to a Michaelis-Menten curve with *GraphPad Prism*. **f** Carnitine concentration-response curves and apparent affinities were determined for hOCTN2 mutant transporters. L-Carnitine concentrations ranging from 1 to 3000 μM were applied to oocytes expressing hOCTN2 mutants of interest. Current responses were measured at −60 mV and fitted to a Michaelis−Menten curve with *GraphPad Prism*. Replicates were measured in 5 oocytes (*n* = 5) across at least 2 batches of oocytes. Error bars represent SEM.

above. Both cryo-EM samples of OCTN2 contain 150 mM NaCl in the buffer. The residues that coordinate Na⁺ are unchanged in position, however the density for Na⁺ is slightly weaker, and there is no density for the axial (basal) water molecule as described for the occluded conformation. This may be due to a number (or combination) of factors e.g. slightly lower resolution of this dataset, the Na⁺ occupancy, or greater protein flexibility of the substrate-free conformation. The cryo-EM condition used here also contains 150 mM NaCl, which is higher than the typical cytoplasmic Na⁺ concentration. Curiously, the Na⁺ cavity remains occluded, without an obvious exit pathway or solvent

accessibility. We therefore designate this as an inward-facing Na⁺-bound conformation, and suggest that further, subtle structural rearrangement is required for Na⁺ release. Similar states have been resolved for the Na⁺ driven MFS MelB[41] as well as SLC13/DASS type transporters[42].

## Voltage-clamp electrophysiology verifies that the Na⁺- and carnitine-binding sites are coupled
To further explore the carnitine and Na⁺ binding sites, wildtype and mutant OCTN2 transporters were expressed in *Xenopus laevis* oocytes

and two-electrode voltage-clamp electrophysiology was utilised to demonstrate the transport of carnitine is Na$^+$-dependent. The application of L-carnitine in the presence of a physiological 100 mM Na$^+$ buffer to oocytes expressing human OCTN2 (hOCTN2) generates an inward current (Fig. 2d). Carnitine is a zwitterion, with no overall charge at pH 7.4, and so the current observed is likely due to the coupling of another charged species. When Na$^+$ is substituted for another monovalent cation caesium (Cs$^+$), no current is observed upon the addition of carnitine, demonstrating the Na$^+$ dependence of carnitine transport. In line with previous work[6,23], wildtype hOCTN2 has an apparent $K_M$ for carnitine in the low micromolar range of 12.0 ± 3.0 μM (at 100 mM external [Na$^+$]). The typical plasma carnitine concentration is ~40–60 μM[26,27], indicating that hOCTN2 operates close to saturation under physiological conditions, which would likely maintain steady carnitine uptake into the cell.

The affinity for Na$^+$ was measured at a saturating carnitine dose (100 μM), with Cs$^+$ used to maintain osmolarity. Fitting these data to the Hill equation gave an apparent affinity for Na$^+$ in the low mM range (8.62 ± 1.8 mM) (Supplementary Table 3), in line with previous studies[6,22,43,44], with one report indicating a lower apparent $K_M$ of 0.3 mM[23], which may reflect the choice of N-methyl-D-glucamine as a counter-ion versus Cs$^+$. Nevertheless, our data supports hOCTN2 having a relatively high affinity for Na$^+$, which in the context of physiological Na$^+$ concentrations suggests the transporter operates close to saturation. In addition, the affinity of carnitine for OCTN2 increases as the inwardly directed Na$^+$ gradient increases (Fig. 2e). A plot of log carnitine $K_M$ against log sodium concentration is fitted by a straight line with a slope close to 1 (Supplementary Fig. 7) supporting a 1:1 stoichiometry of carnitine-Na$^+$ symport, which is in agreement with our occluded structure and previous functional studies on OCTN2[6,22,23,43,44].

Mutation of Na$^+$-coordinating residues resulted in decreases in carnitine affinity; N32A showed a ~18-fold increase in the $K_M$ for carnitine and N210A a ~11-fold increase (Fig. 2f) providing strong evidence that Na$^+$ binding is allosterically coupled to carnitine binding. Variants at these positions are implicated in SPCD. N32S, which is the characteristic mutation in the Faroe Islands (where the incidence of SPCD is 1:300) and has a strong association with sudden death in untreated SPCD[45]. The mechanism through which they are coupled is not obvious from our structures alone, though we hypothesise that the two glutamine residues previously identified as important for sodium-dependent transport[46], Q180 and Q207, play a role in linking the Na$^+$ and carnitine sites. In mouse OCTN3/SLC22A21, a sodium-independent carnitine transporter absent in humans[47,48], the dual glutamine motif is replaced with a dual histidine motif, which is the only significant difference in residues between the carnitine/sodium sites of hOCTN2 and mOCTN3 (78.7% identity, Supplementary Fig. 6). Notably, mutations Q180H and Q207H to hOCTN2 abrogate the sodium sensitivity and the $K_M$ for carnitine remains the same with and without sodium[46], supporting the significance of this dual glutamine motif. These glutamine residues in our hOCTN2 pack against each other and bridge the Na$^+$ and carnitine sites (Fig. 2b). Q180 (TM3) is directly above the Na$^+$ ion and engaged in a hydrogen bond with the sodium-coordinating N210 (TM4) while also forming an inter-helical interaction to TM1 containing N32. Q207 (TM4) coordinates carnitine via a water molecule, while also forming a hydrogen bond from its backbone carbonyl to the sidechain amine group of N32, connecting the carnitine and Na$^+$-binding sites.

The sodium-coordinating residues S28, N32, N210 and the putative coupling residues Q180, Q207 are conserved in human OCTN1, consistent with its function as a Na$^+$-dependent ergothioneine transporter[13]. These residues are absent in the sequences of the other human Na$^+$-independent OCTs and OATs surveyed (Supplementary Fig. 8). Alphafold3[36] predictions support a very similar OCTN1 sodium-binding site to OCTN2 (Supplementary Fig. 6). Although we identified a

likely link between the Na$^+$ and carnitine sites, further structural and functional investigations will be necessary to define the precise details of how coupling between the Na$^+$ binding and structural change at the carnitine-binding site is mediated.

Mutation of key residues in the carnitine-binding site, R471A and Y358A resulted in no measurable currents, making further analysis untenable (Supplementary Table 3, Supplementary Fig. 9). Variants at these sites, specifically R471C/H/P and Y358N, are implicated in the carnitine disorder SPCD[49]. R471C has been shown to localise to the membrane and has ~15% activity of wildtype OCTN2 in CHO cells, which is consistent with the observed reduction in currents for R471A, meanwhile Y358N in CHO cells has ~8% activity, though localisation data for Y358N is unavailable[11,50]. These data are in accordance with the occluded structure, where R471 and Y358 bind the opposing ends of carnitine. We identified Y211 as another residue of interest as it forms part of the carnitine binding site and is one residue downstream Na$^+$-coordinating N210 on TM4. Mutating this residue to an alanine shows a similar increase in carnitine $K_M$ as the mutants above (Fig. 2f). Although the $K_M$ for carnitine was increased, the $K_M$ for Na$^+$ was similar (WT; 8.6 ± 1.8 mM vs Y211A; 14.7 ± 5.3 mM), suggesting that Y211 does not play a substantial role in coupling between the Na$^+$ and carnitine sites. As described above, Y211 plays multiple roles in the substrate binding site, forming a large part of the pocket surface and also binding solvent. Previous studies identified that mutation of Y211 to phenylalanine resulted in reduced carnitine transport whereas the transport of the cationic probe compound tetraethylammonium was unaffected[16], indicating that Y211, through its hydroxyl moiety, plays a role in recognising the carboxyl of carnitine. Together with our structural data, these observations support our proposal that Y211-water interactions are important for the transport of carnitine. Furthermore, we note Y211C/H are variants associated with SPCD—changes that likely causes large structural perturbation of the carnitine-binding site beyond simply affecting solvent structure.

## Gating residues involved in the occluded-to-inward conformational transition of OCTN2

Alternating-access transport by secondary active transporters requires the coordination of gates that enclose or preclude access to the substrate-binding site from either side of the membrane[51]. For MFS fold transporters, the intracellular gates are typically formed from TM4 and TM5 packing against TM10 and TM11, and the extracellular gates are formed by TM1 and TM2 packing against TM7 and TM8 that are symmetry related pairs of transmembrane helices[52].

Structural superposition of the occluded and inward-facing conformations (Fig. 3a) confirms that OCTN2 undergoes large rocker-switch like rearrangements, mediated by a relatively rigid N-terminal bundle together with a more flexible C-terminal bundle. In the inward-facing conformation, the interbundle salt bridges observed in the occluded conformation are broken. These salt bridges contribute to the intracellular gate between the N- and C-terminal bundles, and are conserved in both OCTs and OATs. In OCTN2, R459 (TM11) and E220 (TM4) form a salt bridge in the occluded conformation, that is broken in the inward-facing conformation. In both conformations, E220 is engaged in an ionic interaction with R169 (TM3) that, in turn, coordinates D165 (TM2). Together these interactions link TMs 2,3 and 4 (Fig. 3b). R169Q, R169P and R169W are all variants associated with SPCD[11], all of which we hypothesise affect the positioning of both D165 and E220, and subsequently the gating interaction between R459 and E220. It is possible that these variants have impaired folding and localisation, although R169W has previously been shown to localise normally to the membrane[53]. The R459 gating interaction with E220 appears to stabilise a bent conformation of TM11 (housing the critical carnitine-binding R471). We speculate that this interaction may further support a subtle conformational change in TM4, which contains both

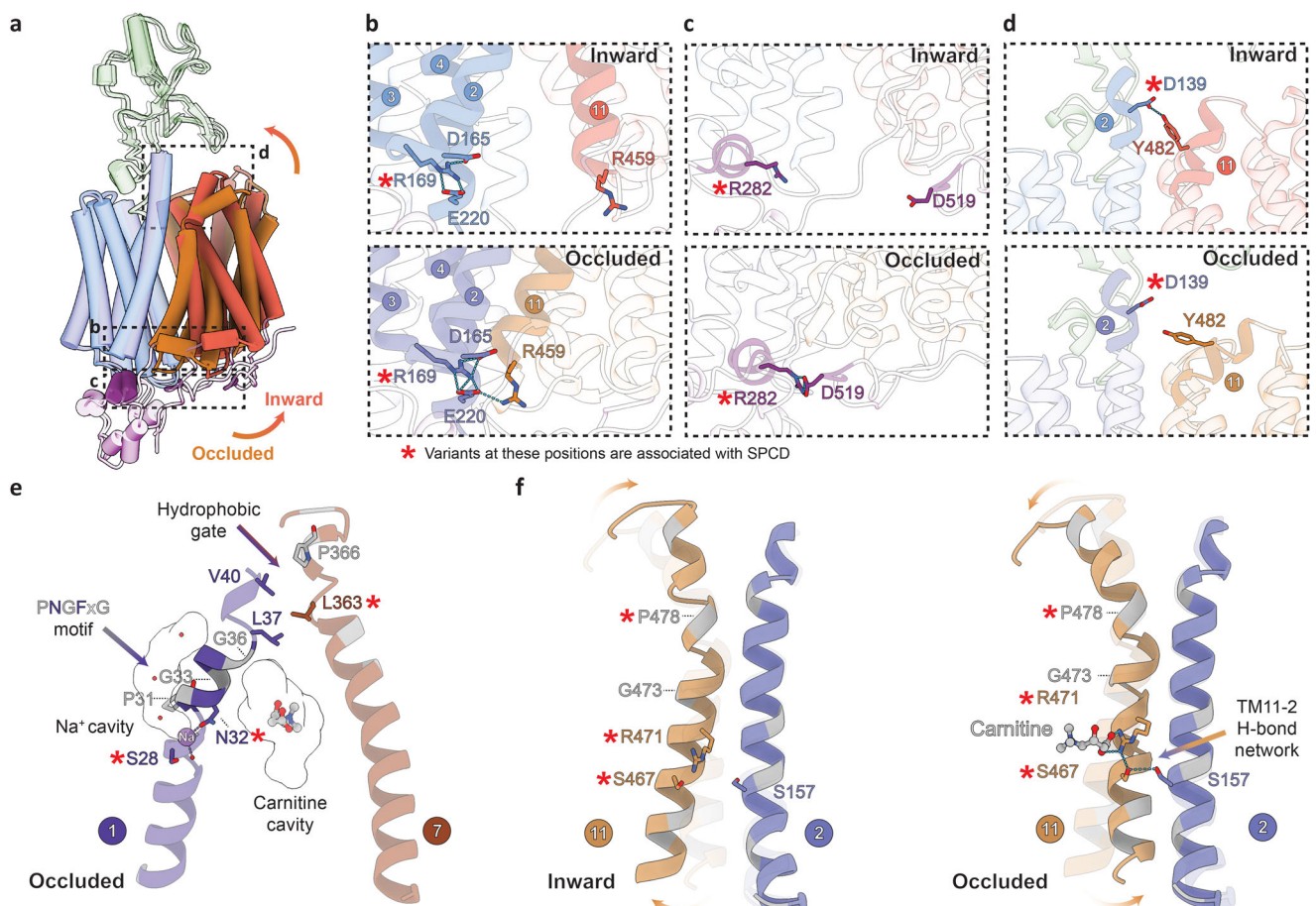

**Fig. 3 | Structural transitions show gating rearrangements and dual function helices. a** Structural superposition of the inward-facing and occluded conformations. The TMs of the inward-facing (IFS) structure are coloured red and light blue, while the TMs of the occluded structure are coloured orange and light purple. The N-terminal bundle is coloured blue/purple and the C-terminal bundle is coloured orange/red. Dashed boxes correspond to views in panels (**b**–**d**). **b** Intracellular gate interactions. In the IFS (top), a charge network is observed in the N-terminal bundle, and in the occluded conformation (bottom), R459 (TM11) interacts with this network. **c** Gating interactions at the intracellular helix (ICH) domain where a salt bridge forms between R282 and D519 in the occluded conformation. Red asterisks indicate positions of variants associated with SPCD. **d** Extracellular gate interactions, where Y482 and D139 H-bond in the IFS (top) and move further apart in the occluded conformation (bottom). Cryo-EM density for residues detailed in (**b**–**d**) is shown in Supplementary Fig. 5. **e** Residues on TM1 (purple) and TM7 (brown) form a hydrophobic extracellular gate above the carnitine-binding site. As well as gating, residues on TM1 form the Na⁺-binding site. The PNGFxG motif observed in the Na⁺-dependent hOCTN2, hOCTN1 and mOCTN3 is indicated. Proline and glycine residues are coloured grey. **f** Flexibility of TM11 (orange) and formation of a hydrogen-bonding network with TM2 (blue) in the occluded carnitine-bound structure. Arrows and transparent overlay illustrates TM11 bending during the inward-occluded conformational change. The position of the conserved G473/P478 motif is indicated.

carnitine and Na⁺-binding residues. Another latching interaction occurs between the N-terminal and C-terminal portions of the ICH, where R282 (N-terminal) form a salt bridge with D519 of the (C-terminal, Fig. 3c). This interaction was also observed in OCT1[29], and we note R282 is a part of a "PESPR" motif conserved in OCTs and OATs and found in glucose transporters[54]. R282Q is a variant associated with SPCD, and localises normally to the plasma membrane[53]. Altogether, these data support a pathogenicity mechanism where variants can disrupt transporter gating.

The inward-facing structure shows a tightening of the interactions between the residues involved in the extracellular gates (Fig. 3c). Specifically, D139 (TM2) forms a hydrogen bond with Y482 (TM11) in the inward-facing structure, stabilising a straightened conformation of TM11 (Fig. 3d). The D139-Y482 hydrogen bond is not present in the occluded structure. In OCT1, the equivalent interaction is a salt bridge between R486_OCT1 and D149_OCT1, which similarly stabilises the inward-facing conformation. For OCTN2, D139N is a variant potentially associated with SPCD that does localise correctly to the membrane, but does not impair transport activity in uptake assays[11]. It may be that an asparagine can function similarly in a gating capacity to the aspartate,

and that association of this variant to SPCD involves a more complex mechanism beyond substrate transport alone.

We identify the conformationally flexible TM1 as a key helix with dual function, as residues on this helix contribute to both Na⁺-binding and extracellular gating formation. We observe a hydrophobic extracellular gate directly above the substrate-binding site, similar to that seen in OCT1, where residues of gating helices TM1 and TM7 pack together in both the occluded and inward conformations (e.g. L37 and V40 on TM1 packing against L363 and P366 on TM7). Notably, the Na⁺-coordinating residues and the gating residues are separated by a helical break near G36 on TM1 and we speculate that Na⁺-binding may stabilise a conformation in which the extracellular gate can form. G36 is part of a proline/glycine motif, PNGFxG, that includes the Na⁺-binding residues N32. This motif is present in the Na⁺-driven hOCTN1 and mOCTN3, but notably absent in OCTs and OATs, suggesting a possible OCTN-specific role of this motif. As mentioned previously, N32 is also engaged in an inter-helical hydrogen bond with Q207 of TM4 (Fig. 2a) and Na⁺ binding appears to help staple these gating helices together. The conformational dynamics of TM1 and how they relate to both coupling and gating requires further investigation.

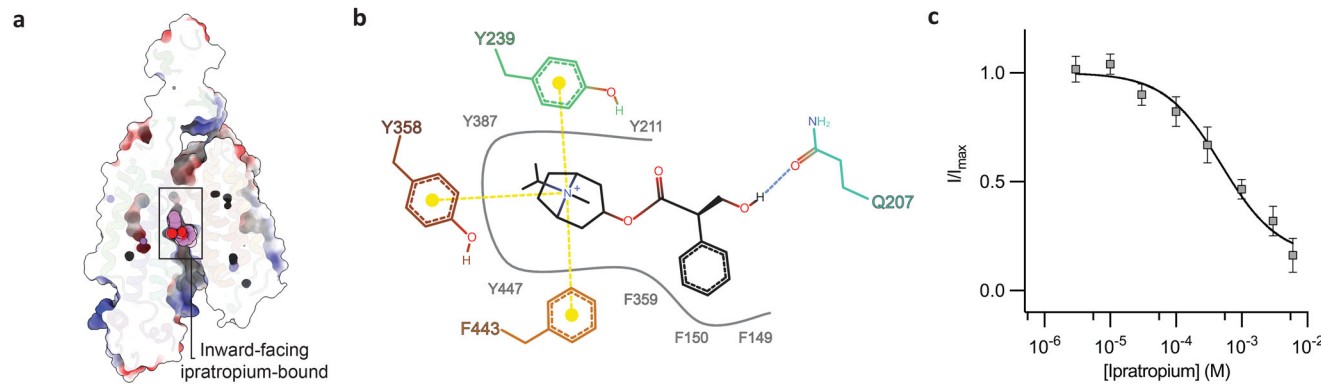

**Fig. 4 | Inward-facing ipratropium-bound OCTN2. a** Cutaway shows ipratropium bound within the solvent-accessible cavity in the inward-facing conformation. **b** Representation of ipratropium-OCTN2 interactions, showing cation- interactions between the aromatic cage residues and the quaternary ammonium group on the tropane ring, along with a weak hydrogen bond between Q207 and the ipratropium hydroxyl group. **c** Ipratropium inhibition response curves were measured by competing increasing concentrations of ipratropium (3–6000 μM) against carnitine at the approximate $EC_{50}$ of 10 μM. The current measured at −60 mV was fitted to a three parameter inhibitor dose response with *Graphpad Prism*. Replicates were measured in five oocytes ($n = 5$) across at least two batches of oocytes. Error bars represent ±SEM.

The conformational flexibility of TM11 appears to be linked with substrate recognition, with R471 (responsible for carnitine coordination) located near the midpoint of this helix, between the two segments that bend during the transition between inward-facing and occluded conformations. Specifically, TM11 bends about a glycine/proline motif G473 and P478, positioned just above R471 (Fig. 3f). A variant here, P478L, is associated with SPCD[55], and in human OCT1, the equivalent residues are also implicated in the inward-outward conformational transition[29]. Cell uptake assays demonstrated that P478L impaired carnitine uptake, did not change $Na^+$ activation kinetics and enhanced TEA uptake[16], altogether supporting a link between TM11 flexibility and carnitine binding. In the OCTN2 occluded conformation, R471 is involved in a hydrogen bonding interaction with S467, coordinating both the hydroxyl and the backbone carbonyl at the point where the lower segment of TM11 kinks toward TM2 in the occluded conformation. S467 (TM11) in turn makes an interhelical hydrogen bond with S157 (TM2) and forms an interaction network linking the N- and C- terminal bundles (Fig. 3e). Furthermore, we note that S467C is a variant associated with SPCD[56] that has been shown to increase the $K_M$ for carnitine in cell uptake assays -15-fold[57]. The equivalent arginine-serine network in *Rattus norvegicus* OAT1 has been implicated in gating dynamics and the allosteric modulation of OAT1 by chloride[25], with chloride binding to $R466_{rOAT1}$ in the inward-facing conformation stabilising this network. The H-bond network is only observed for OCTN2 in the occluded conformation and is absent in the inward-facing conformation, supporting that the network helps to stabilise a kinked conformation of TM11 that packs against TM2. In human URAT1/SLC22A12 (another OAT family member), the equivalent arginine, $R477_{URAT1}$, has a direct role in substrate coordination, forming a hydrogen bond with the substrate urate. This position is essential for activity and further thought to be key for coupling substrate binding to the outward-inward transition[58] and we suggest R471 plays a similar role in OCTN2.

### The ipratropium-bound structure demonstrates how OCTN2 binds positively charged drugs

We selected the anticholinergic bronchodilator, ipratropium, as a model compound to investigate how OCTN2 binds positively charged drug molecules. We determined the cryo-EM structure of OCTN2 in the presence of ipratropium in an inward-facing conformation to 3.0 Å resolution. Strong density for ipratropium is present at the substrate-binding site (Fig. 1b), which is fully solvent-accessible. The ammonium group of ipratropium is similarly positioned to where the ammonium group of carnitine is in the occluded conformation, and forms cation-π

interactions with the aforementioned hydrophobic residues. The phenyl group of ipratropium tucks into a hydrophobic portion of the pocket formed by F149, F150 and F359. There are few polar interactions, with only one weak hydrogen bond between Q207 and the hydroxyl group of ipratropium, and notably, no direct interactions with R471. Overall, the structure is highly similar to the inward-facing conformation, and again, there is density that we attribute to $Na^+$ at the same binding site, albeit slightly weaker in intensity (Supplementary Fig. 3). The distinct lack of polar interactions or engagement of R471 are in stark contrast to the interactions observed in the occluded structure.

It is notable that in the cryo-EM conditions here, ipratropium did not induce occlusion like with the native substrate carnitine. A structural overlay of the ipratropium-bound structure and the occluded structure demonstrates that ipratropium in principle could fit within the occluded substrate-binding site cavity (Supplementary Fig. 10), with only minor adjustments to the substrate-binding site to alleviate a steric clash between the tropane moiety and F443 of TM10. We propose that the absence of the R471 interaction with ipratropium likely accounts for the lack of occlusion. Overall, there are only very subtle differences at the residue level between the inward-facing substrate free and the ipratropium-bound structures. The density indicates some flexibility at R471 in both the inward-facing and ipratropium-bound structures, but both are pointing down towards the extracellular side and wedged between S157 and S467, preventing the H-bond network we observe in the occluded conformation (Fig. 3e). It is possible that the binding mode for ipratropium to the outward-facing conformation may be different to that observed here.

These structural insights are mirrored in our functional data. Ipratropium did not elicit any current when applied to oocytes expressing hOCTN2, and rather acts as an inhibitor of carnitine transport, with an $IC_{50}$ in the μM range ($615 \pm 174.5$ μM, $n = 5$) (Fig. 4). Given that ipratropium bears a positive charge, even uncoupled transport (i.e. without sodium) should result in the generation of current. While these data do not rule out slow or electroneutral transport, they strongly support that ipratropium primarily binds and blocks the transporter, rather than being efficiently translocated. This is consistent with a mechanism where the R471-carboxylate interaction is required to couple substrate binding to transport.

### Discussion

Our cryo-EM structures of OCTN2 provide structural insight into the mechanism of this physiologically significant transporter. The occluded structure defines the substrate and $Na^+$-binding sites and reveals

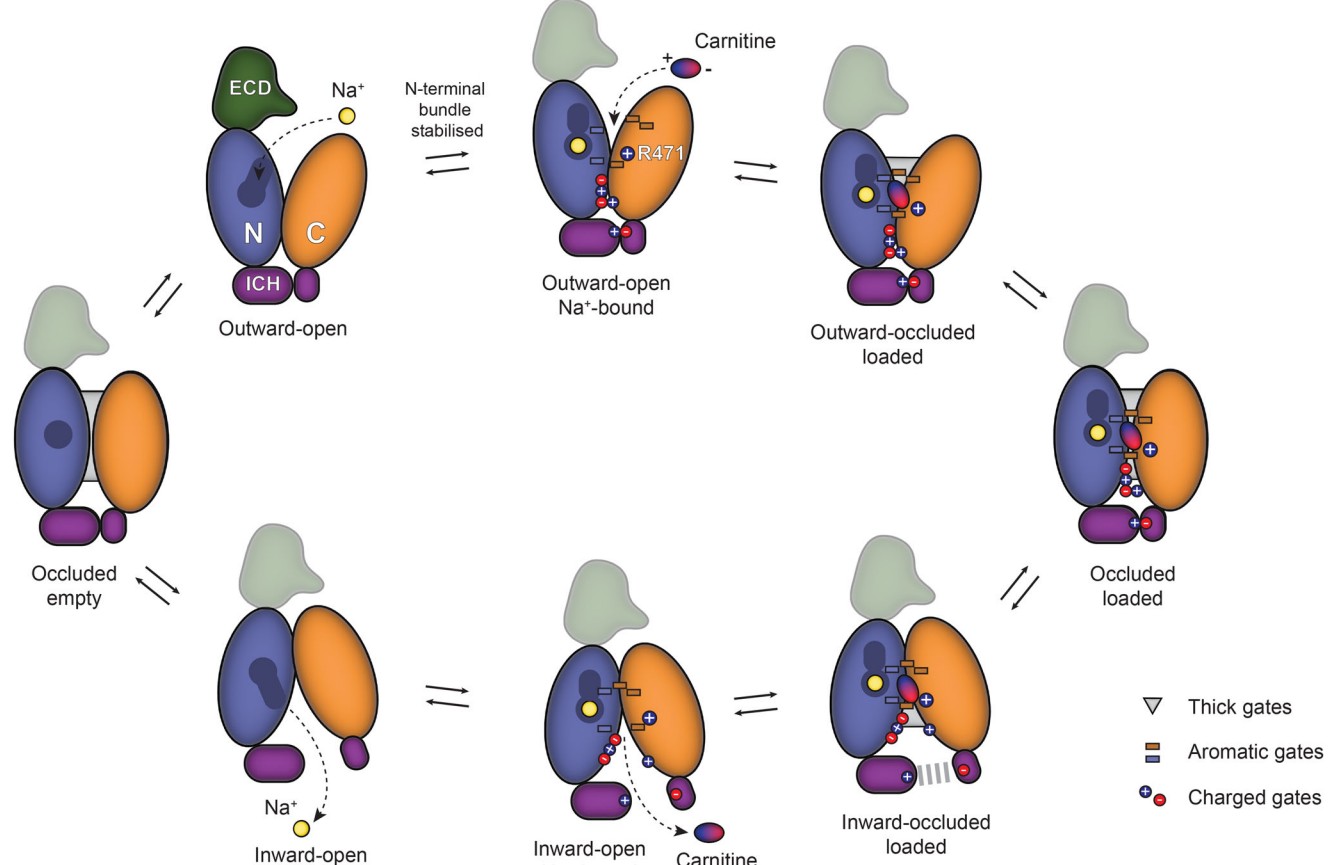

**Fig. 5 | Proposed OCTN2 Na⁺-coupled carnitine transport cycle.** We propose that OCTN2 follows a variation of the MFS-type transport cycle, in which Na⁺-binding within the N-terminal bundle (blue) allosterically modulates the transporter. We hypothesise that Na⁺ binds first (similar to other Na⁺-solute symporters[41,42]) and this primes the transporter for subsequent carnitine binding, which involves the critical R471 within the C-terminal bundle (orange). Na⁺-binding is likely to influence the conformational dynamics associated with extracellular gating, although kinetic evidence for the binding order remains to be established. Like other OCT and OAT transporters, electrostatic interactions (circles) govern the intracellular gate and aromatic gating residues (rectangles) that surround the binding pocket (effectively "thin" gates). The role of the extracellular domain (ECD; green) in the transport cycle is unclear, as the structural data here does not support significant interaction with the C-terminal bundle.

key molecular interactions that underpin high affinity carnitine transport. Functional analysis of OCTN2 shows that carnitine transport is electrogenic and Na⁺-dependent, suggesting Na⁺-coupled carnitine transport and supporting our assignment of an allosteric Na⁺-binding site. To our knowledge, this Na⁺-binding site has yet to be observed in MFS-type transporters in terms of location and architecture. We observe that the distinct Na⁺ and carnitine cavities are bridged by a dual glutamine motif, previously implicated in Na⁺ activation kinetics[46]. The occluded aqueous cavity housing the Na⁺-binding site is unusual, and we hypothesise that Na⁺ binding and ordering of solvent within this cavity may stabilise the N-terminal domain and contribute to transporter gating. From both a structural and functional perspective, OCTN2 is a marked departure from other members of the SLC22 family, and presents the transport mechanism of the OCTN sub-family (Fig. 5).

Analysis of these structures provide context for how a number of variants associated with SPCD affect the transport of carnitine. Some variants have clear structural explanations for their loss-of-function, for example, variants can disrupt binding of carnitine (R471C/H/P), affect the Na⁺-binding site (N32S), disrupt the TM2-TM11 H-bonding network (S467C) or perturb gating residues conserved in OCTs/OATs (R169Q/P/W). However, variant interpretation is complicated by factors such as membrane targeting: for instance, a serine mutation at one of the other Na⁺-binding residues, N210, causes a reduction in uptake, but also shows mixed localisation,[11] which may itself contribute to

disease. Nonetheless, the structural data presented here, in particular the occluded structure, provide a valuable framework for interpreting the functional effects of variants. These data may further support classification for diagnosis and clinical management and we note that the machine learning model engineered for this purpose did not contain explicit information about the binding of either Na⁺ or carnitine[11].

While a number of SLC22 family members contribute significantly to the pharmacokinetics of a number of drugs and xenobiotics[12,59], it is thought that comparatively, OCTN2 is less of a general drug transporter and more a specialised nutrient transporter[18]. Our inward-facing structure with the anticholinergic ipratropium demonstrates how OCTN2 can bind compounds bearing a quaternary ammonium group. However, we did not observe transport of ipratropium and rather saw inhibition of carnitine transport. We suggest that this is because ipratropium is able to engage the OCTN2 aromatic cage, but is unable to form a salt bridge with R471. Indeed, OCTN2 has been shown to transport TEA in a Na⁺-independent manner[15], albeit much less efficiently than carnitine (0.2-3% transport rate relative to carnitine)[18,23]. Similar to ipratropium, cationic TEA would also not be able to form a salt bridge with R471, further supporting that the R471 salt bridge is required for efficient transport by OCTN2.

Although a full understanding of the mechanism of OCTN2 transport will require additional structural characterisation, particularly of outward-facing conformations, the data presented here

provides a strong foundation for our proposed mechanism. Collectively, our findings reinforce that OCTN2 functions as a highly selective, high affinity transporter, mediating carnitine uptake via a Na$^+$-dependent mechanism that, to our knowledge, has not been observed in the SLC22 family or in MFS transporters more broadly.

# Methods

## OCTN2 protein expression and purification

Full-length human wild-type OCTN2 (Uniprot accession: O76082) was synthesized and cloned into the pcDNA3.1 vector by Genscript. The gene was subcloned into the BacMam vector[60], with a C-terminal eGFP and PreScission cleavage site, using primers (Fwd: CCACTCCCAGTT-CAATTACAGCTCTTAAGGGGCCACCATGCGGGACTACGACGAGG and Rev: CAGCACTTCCAATCTAGATTCGAAAGCGGCGCCGAAGGCTGTGC TTTTAAGGATT. IDT). Baculovirus was generated in MAX Efficiency DH10B cells using the manufacturer's protocol (ThermoFisher) and amplified to P2 in Sf9 cells. OCTN2 was expressed in HEK cells, similarly to previous work with OCT1[29], except the Freestyle 293-F cell line (ThermoFisher; R79007) was used instead of the GNTI- cell line. Fetal bovine serum (ThermoFisher) was added to the Freestyle media to a final concentration of 2% v/v. After 24 h 10 mM sodium butyrate was added, and the temperature lowered to 30 °C for a further 24 h. Cells were harvested by centrifugation (5500 × $g$) and flash frozen with liquid N$_2$. For purification, 20–30 g of cells were resuspended in lysis buffer comprised of 2xTBS (40 mM Tris-HCl pH 8.0, 300 mM NaCl) and supplemented with 10 mM benzamidine, 10 mM 6-aminocaproic acid and 1 mM PMSF. Cells were lysed by sonication with 2 ×25 s pulses (0.9 s on and 0.1 s off). Cellular debris was removed by centrifugation at 10,000 × $g$ for 20 min and the membranes harvested by centrifugation at 100,000 × $g$ for 1 h. Membranes were resuspended in 2xTBS pH 8.0, 2% DDM, 0.2% CHS to a total volume of 100 mL and solubilised with gentle stirring for 1 h at 4 °C. Insoluble material was removed by centrifugation at 100,000 × $g$ for 30 min. The supernatant was then incubated with 4 mL freshly prepared GFP-nanobody conjugated resin for 1 h. Resin was prepared by conjugating ~20 mg of GFP-nanobody to 4 mL swollen CNBr-activated Sepharose 4B (Cytiva). The expression and purification of GFP-nanobody was performed as previously described[28]. Following sample application at 4 °C, the resin was then washed with 25 column volumes 2xTBS supplemented with 0.02% v/v GDN (Anatrace). Bound OCTN2 was then cleaved from the resin overnight using PreScission protease (~1:10 protease to OCTN2), in 8 mL of 2xTBS, 1 mM EDTA and 1 mM DTT. The eluted protein was concentrated using an Amicon 100 kDa MWCO Ultra spin column (Millipore) and then subjected to size-exclusion chromatography using a Superose 6 increase 10/300 GL column (Cytiva) equilibrated with 1xTBS, 0.02% GDN. Purity was assessed using SDS-PAGE and the fractions corresponding to OCTN2 were pooled and concentrated to 6–7 mg/mL.

## Cryo-EM grid preparation

For grid preparation, 3 uL of 3 mg/mL purified OCTN2 was applied to a freshly glow discharged holey gold grid, with UltrAuFoil R0.6/1, 300 mesh used for inward-facing, and UltrAuFoil R1.2/1.3 300 mesh grids used for OCTN2 occluded and OCTN2 ipratropium. Grids were blotted for 4 s at 22 °C, 100% humidity and plunge cooled in liquid ethane using a Vitrobot Mark IV (Thermo Fisher). L-carnitine (Sigma-Aldrich) was prepared in SEC buffer and added to a final concentration of 10 mM. Ipratropium (Sigma-Alrich) was prepared in DMSO and added to a final concentration of 1 mM (final DMSO concentration of 0.5%). The protein and ligand mixtures were incubated on ice for 2 h before vitrification.

## Cryo-EM data collection

Grids were screened for ice thickness and particle density using a ThermoFisher Talos Arctica transmission electron microscope (TEM)

at 200 kV. Structure determination was carried out using a ThermoFisher Titan Krios TEM operating at 300 kV equipped with a Gatan BioQuantum energy filter (with a 15 eV slit width for OCTN2 inward, OCTN2 ipratropium and a 25 eV width for OCTN2 carnitine), and a Gatan K3 camera. Automated data acquisition was performed using *EPU* v 3.4, with a defocus range of −1 to −1.5 μm for both OCTN2-inward and OCTN2-ipratropium, and −0.5 to −2.0 μm for OCTN2-carnitine. For OCTN2-inward, 16,735 movies were collected and at a magnification of 105,000× with a pixel size of 0.83 Å. A total exposure of 81 electrons per Å$^2$ spread over 105 frames was used for OCTN2-inward, with an exposure time of 8.4 s. For OCTN2 ipratropium, a total exposure of 79 electrons per Å$^2$ spread over 100 frames, with an exposure time of 8 s. For OCTN2-carnitine, 22,422 movies were collected at the aforementioned magnification and pixel size, with a total exposure of 82 electrons per A$^2$ spread over 100 frames with an exposure time of 8 s.

## Cryo-EM data processing

All image processing and map reconstruction was performed using *Cryosparc* v 4.6.0. The processing workflow for the cryo-EM structures are summarised in Supplementary Fig. 1. Micrographs were processed using the Patch Motion Correction and Patch CTF Estimation jobs, then manually curated to remove micrographs with CTF fit resolution cutoffs greater than 5 Å. Particles were picked initially using the blob picker on a subset of micrographs to generate templates, after which template picker was used for each dataset. Particles were extracted with a box size of 320 pixels and Fourier cropped to 160 pixels, and then subjected to multiple rounds of 2 d classification. For all datasets, selected particles were then used in multiple rounds of reference-free ab initio reconstruction (2–5 classes, max resolution = 6 Å), followed by heterogeneous refinement. Classes were then subjected to non-uniform (NU) refinement (initial lowpass resolution = 15 Å), followed by local refinement masking out the density corresponding to the micelle. Masks were constructed using *ChimeraX* and *Cryosparc* was used to dilate and add a soft edge. 3d classification without alignment was used to remove empty micelles or "junk" particles. Final rounds of refinement were performed with uncropped particles.

## Model building

Models were built into cryo-EM density using a combination of *coot* v 0.9.8.94 and the *ISOLDE* v 1.6 plugin in *ChimeraX* v 1.9. Initial models were acquired from the Alphafold database, and first fit to the maps using *ISOLDE*. Manual corrections to the model were made using both *ISOLDE* and *coot*, with rounds of automated real space refinement performed using *phenix* v 1.21.1_5286 *real space refine*.

## Preparation of hOCTN2 mRNA

The gene for hOCTN2 (excluding purification tags) from the BacMam vector was subcloned into pOTV and point mutations of interest introduced via site-directed mutagenesis by Genscript™. The plasmid DNA product was transformed into NEB® 5- alpha competent *Escherichia coli* cells (New England BioLabs Inc), following manufacturer's instructions then purified using a Midiprep Kit (QIAGEN®). The plasmid DNA was linearised with the restriction enzyme Spe1 (New England BioLabs, Inc) and transcribed into mRNA by T7 RNA polymerase with the AMBION mMESSAGE mMACHINE™ T7 transcription kit (Invitrogen™, Thermo Fisher Scientific).

## Preparation of Xenopus laevis oocytes

Oocytes were defolliculated by agitation with 2 mg/mL collagenase A (Roche) at 18 °C for 30 min. Isolated oocytes were microinjected with 8 ng hOCTN2 mRNA. Injection needles were made with 3.5' Drummond glass capillaries (Drummond Scientific Company) using a PC-100 microelectrode puller (Narishige). Injected oocytes were stored at

18 °C in ND96 storage buffer (96 mM NaCl, 2 mM KCl, 1 mM $MgCl_2$, 1.8 mM $CaCl_2$, 5 mM hemisodium-HEPES, pH 7.4 supplemented with 50 μg/ml gentamycin, 2.5 mM sodium pyruvate, 50 μg/mL tetracycline and 0.5 mM theophylline) for three days to allow the expression of hOCTN2 on the oocyte plasma membrane.

### Electrophysiology studies

Current recordings were measured using two-electrode voltage clamp electrophysiology with an AxoClamp 900A microelectrode amplifier (Axon Instruments) and a PowerLab 4/26 chart recorder (ADInstruments) interfaced with *LabChart* software (version 8) (ADInstruments). Current-voltage (IV) relationships were measured by applying voltage steps with a Digidata 1550B with Humsilencer (Axon Instruments) controlled by an IBM-compatible computer with *Axon*™ *pClamp11*™ software (Molecular Devices). L-carnitine (Sigma Aldrich) and ipratropium bromide (Sigma Aldrich), were dissolved in ND96 buffer and applied to oocytes through perfusion into the recording bath where they were held and voltage clamped at −30 mV. The recording bath was grounded using a salt bridge containing 3 M KCl and 2% agarose gel which was connected to a 3 M KCl reservoir to minimise offset potentials. Voltage pulses at 10 mV intervals between −100 mV to +60 mV were applied to oocytes every 150 milliseconds to generate IV plots. To determine the current activated by each compound, baseline IVs were measured in the buffer alone immediately prior to compound application and subtracted from the compound-elicited IV. To perform $Na^+$ dependence assays, carnitine doses were applied to oocytes expressing hOCTN2, and related mutants, in the presence of $Cl^-$ buffers of differing $Na^+$ concentration (10–100 mM NaCl, 100 mM - [NaCl] CsCl, 2 mM KCl, 1 mM $MgCl_2$, 1.8 mM $CaCl_2$, 5 mM HEPES, buffered to pH 7.4 with Tris base). $Cs^+$ was used as the substituent cation in these experiments. In $Na^+$ titration experiments, saturating concentrations of carnitine were applied to oocytes expressing hOCTN2 and mutants (1–150 mM NaCl, 150 mM - [NaCl] CsCl, 2 mM KCl, 1 mM $MgCl_2$, 1.8 mM $CaCl_2$, 5 mM HEPES, buffered to pH 7.4 with Tris base).

### Ethics statement

*Xenopus laevis* oocytes were supplied by the Victor Chang Cardiac Research Institute. Stage V oocytes were obtained by surgical laparotomy from female *Xenopus laevis* frogs. Surgical procedures were approved by the Garvan Institute/St Vincent's Hospital Animal Ethics Committee (Animal Research Authority 23_11) under the Australian Code of Practice for the Care and Use of Animals for Scientific Purposes.

### Reporting summary

Further information on research design is available in the Nature Portfolio Reporting Summary linked to this article.

## Data availability

The cryo-EM maps have been deposited in the Electron Microscopy Data Bank (EMDB) under the accession codes EMD-71735 (OCTN2 inward-facing), EMD-71540 (OCTN2 occluded) and EMD-71597 (OCTN2 ipratropium). Atomic coordinates have been deposited in the Protein Data Bank (PDB) under accession codes 9PMD (OCTN2 inward-facing), 9PDQ (OCTN2 occluded) and 9PFB (OCTN2 ipratropium). All raw data for the electrophysiology studies (corresponding to Figs. 2d–f and Fig. 4c) are available in the **Source Data** provided with this paper. Source data are provided with this paper.

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

## Acknowledgements

A.G.S. was supported by a National Health and Medical Research Council (APP2016308). R.M.R was supported by the Australian Government through an Australian Research Council Future Fellowship (FT220100717). We acknowledge the use of the Victor Chang Innovation Centre and the Electron Microscope Unit at UNSW Sydney, funded in part by the NSW Government. We also acknowledge the use of the University of Wollongong Cryogenic Electron Microscopy Facility at Molecular Horizons. This research was conducted by the Australian Research Council Industrial Transformation Training Centre for Cryo-Electron Microscopy of Membrane Proteins for Drug Discovery (IC200100052).

## Author contributions

Y.C.Z. and A.G.S. conceived the study. J.S.D and Y.C.Z. performed protein expression and purification. J.S.D, Y.C.Z., and S.H.J.B. performed the cryo-EM data collection and analysis. C.B. performed electrophysiology.

J.S.D. wrote the initial draft and all authors edited the manuscript. R.M.R. and A.G.S. supervised the study.

## Competing interests

The authors declare no competing interests.
