## [Transparent Peer Review file · Nature Communications]

Structural basis of sodium ion-dependent carnitine transport by OCTN2

Corresponding Author: Dr Alastair Stewart

Version 0:

Reviewer comments:

Reviewer #1

(Remarks to the Author)

The study by Davies et al. reports the first structures of the carnitine transporter OCNT2 (SLC22A5) in multiple states and conformations. Coupled with these structural studies, the authors use mutagenesis and electrophysiology to tease apart the sodium and substrate binding of the transporter. Finally, the authors use their structural and functional results and underlying biophysical insights to begin to explain the transport cycle and the pathogenic mechanisms of OCNT2 mutations that cause systemic primary carnitine deficiency.

The study is well designed, executed, analyzed and presented. The science is unambiguously strong and I have only modest suggestions for the analysis and presentation.

I suggest only minor revisions to improve the clarity of the manuscript and figures, and have no doubts this can be easily revised for publication.

- While the abstract and introduction explicitly mention OCNT2 facilitating fatty acid transport, based on Fig. 1A its role in FA import is indirect. I suggest rewording this slightly to clarify its mechanism and role in FA metabolism and physiology, as an initial reading expects OCNT2 to directly act upon fatty acids.

- Lines 55-56: About carnitine concentrations in tissue, is this meant to be the extracellular microenvironment or the cytoplasmic concentration? Clarifying this wording will help in understanding the transporter function.

- E383, K436, Y211, S231, and G236 are discussed in reasonable detail for their interactions in/near the binding site, but not shown in a figure. Figure panel(s) to illustrate their locations and interactions should be included in Fig. 2 or an extended data figure.

- Two (technical) hypotheses are proposed for the presence of Na in the ligand-free structure: a semi-occupied binding site or detergent/lipid effects. However, the reaction mechanism in Fig. 5 suggests a discrete Na-only bound state. In fact, such sequential binding and allosteric interaction between substrate and coupling ion has been observed for other SLCs, such as the SLC13/DASS family. Through the functional and e-phys studies, is it known if the sodium and substrate binding are sequential for OCNT2?

- On lines 289-290 and 297-298, the locations of pathogenic mutations are noted, but an explicit hypothesis for their mechanism is not provided. These should be commented upon if at all possible, even if tentatively.

- An extended data figure should be provided to show the density of the side chains highlighted in Fig. 3b-d. This is needed to justify or give confidence in the modeling of these side chains on the periphery of the protein, where the map quality is likely lower.

- The text (lines 341-342) describes TM11 bending, while figure 3f appears to show TM11 breaks between R471 and G473. Can this figure be modified, or an overlay provided in an extended data figure, to better illustrate this movement?

- It is difficult to visualize the similarity or differences in the coordination of carnitine and ipratropium. Can the authors provide an overlay of the two substrate-bound structures to better illustrate this point?

- I am confused by the description of ipratropium (page 17), particularly as it relates to possible binding in an occluded conformation. Based on the described structures, it does not trigger the same conformational change as a bona fide substrate. Correspondingly, it does not generate transport currents and blocks carnitine transport currents. Therefore, while the authors do not directly assert it, ipratropium appears to meet all criteria for an inhibitor and the results should be analyzed accordingly.

- The discussion's comments on R471 are confusing and appear contradictory. How can the data from (poor

substrate/inhibitor) ipratropium suggest that engaging this residue is critical for transport, while substrate TEA's inability to interact with this residue also highlights its importance?

- Was heterogeneous refinement used during data processing? As described in the methods, the authors used an unusual procedure by going straight from ab initio to non-uniform refinement.
- Were any other non-junk classes observed during data processing? The maps from ab initio (or heterogeneous refinement) should be included in ED Fig. 1 to give a sense of whether any alternative conformations or species are present in the sample. (Though I appreciate the compact and neat layout of the data processing figure.)

Reviewer #2

(Remarks to the Author)

This report details the structure of a carnitine transporter, OCTN2. The cryo-EM structure was obtained at good resolution in an occluded conformation and shows the substrate binding site, as well as a bound Na⁺ ion. In addition, the structure of an inward-facing, substrate-free state was obtained. Finally, the authors demonstrate the structure of an ipratropium-bound inward-facing conformation. Ipratropium is characterized as an inhibitor of OCTN2, since no transport activity was detected. This is very nice work that has several novel aspects. 1) While the structure of OCTs is known (not surprisingly showing the MFS fold), the structure of a member of the OCTN subclade has not been determined yet. While the fold is as expected, this new structure informs on transport mechanism. For example, Na⁺ is bound in the inward-facing, substrate-free conformation, suggesting sequential Na⁺/substrate binding. 2) The Na⁺ binding site is unusual in the MFS family and is connected to a water-filled cavity. The implications of this finding for the mechanism await further study. 3) The inhibitor-bound structure is novel and is interesting, due to its high similarity to the inward-facing structure. Overall, this is sound work that advances the field. I have some minor comments:

- Is ipratropium a competitive inhibitor? If yes, has this been experimentally shown?
- Is substrate uptake inhibited by ipratropium? Currents don't measure flow, so they are only an indirect metric of transport.
- It would be important to show original OCTN2 current traces in the extended data.

Version 1:

Reviewer comments:

Reviewer #1

(Remarks to the Author)

The study by Davies et al. reports the first structures of the carnitine transporter OCNT2 (SLC22A5) in multiple states and conformations. Coupled with these structural studies, the authors use mutagenesis and electrophysiology to tease apart the sodium and substrate binding of the transporter. Finally, the authors use their structural and functional results and underlying biophysical insights to begin to explain the transport cycle and the pathogenic mechanisms of OCNT2 mutations that cause systemic primary carnitine deficiency.

The authors have done an excellent job addressing my notes, and I have no reservations in recommending this manuscript for publication.

Reviewer #2

(Remarks to the Author)

This a comprehensive revision of the original manuscript, I have no further comments.

We thank the reviewers for their positive and constructive comments, which we have incorporated into our revised manuscript. We have addressed each reviewer comment (shown in *italics*) in our point-by-point response below (response in blue), with changes made to the text **underlined**.

Reviewer #1

The study by Davies et al. reports the first structures of the carnitine transporter OCNT2 (SLC22A5) in multiple states and conformations. Coupled with these structural studies, the authors use mutagenesis and electrophysiology to tease apart the sodium and substrate binding of the transporter. Finally, the authors use their structural and functional results and underlying biophysical insights to begin to explain the transport cycle and the pathogenic mechanisms of OCNT2 mutations that cause systemic primary carnitine deficiency.

The study is well designed, executed, analyzed and presented. The science is unambiguously strong and I have only modest suggestions for the analysis and presentation.

I suggest only minor revisions to improve the clarity of the manuscript and figures, and have no doubts this can be easily revised for publication.

- While the abstract and introduction explicitly mention OCNT2 facilitating fatty acid transport, based on Fig. 1A its role in FA import is indirect. I suggest rewording this slightly to clarify its mechanism and role in FA metabolism and physiology, as an initial reading expects OCNT2 to directly act upon fatty acids.

We have re-worded the abstract accordingly. Lines 3-5 now read:

“The carnitine transporter OCTN2 (novel organic cation transporter 2, SLC22A5) **mediates** carnitine uptake across the plasma membrane **and as such facilitates fatty acid metabolism in most tissues.**”

- Lines 55-56: About carnitine concentrations in tissue, is this meant to be the extracellular microenvironment or the cytoplasmic concentration? Clarifying this wording will help in understanding the transporter function.

This is taken here to reflect the intracellular concentrations of carnitine. We also wish to acknowledge that measurements from tissue homogenates are not a direct measurement of cytoplasmic concentration, and have re-worded accordingly. Lines 54 to 56 now reads:

“Typically, the concentration of carnitine in plasma is in the low μM range (~40-60), while tissue concentrations (**taken here to approximate cytoplasmic concentrations**) are in the low mM range (~1-5 mM)^{26,27}.”

- E383, K436, Y211, S231, and C236 are discussed in reasonable detail for their interactions in/near the binding site, but not shown in a figure. Figure panel(s) to illustrate their locations and interactions should be included in Fig. 2 or an extended data figure.

We have added **Extended Data Figure 5**, which shows the position of these residues in relation to the carnitine-binding site, and also the corresponding cryo-EM density. We have amended the text to refer to this figure.

- Two (technical) hypotheses are proposed for the presence of Na in the ligand-free structure: a semi-occupied binding site or detergent/lipid effects. However, the reaction mechanism in Fig. 5 suggests a discrete Na-only bound state. In fact, such sequential binding and allosteric interaction between substrate and coupling ion has been observed for other SLCs, such as the SLC/DASS family. Through the functional and e-phys studies, is it known if the sodium and substrate binding are sequential for OCNT2?

We thank the reviewer for this insightful comment. We have clarified the text to reflect our hypothesis that the ligand-free Na⁺-bound conformation is a bona fide intermediate in the transport cycle. We have removed the technical hypotheses, and have amended the text as follows (lines 198-201):

“We therefore designate this as an inward-facing Na⁺-bound conformation, and suggest that further, subtle structural rearrangement is required for Na⁺ release. Similar states have been resolved for the Na⁺ driven MFS MelB⁴² as well as SLC13/DASS type transporters⁴³.”

Our electrophysiology data does not resolve microscopic binding order. In the caption of Figure 5, we have acknowledged this on line 430-431: “although kinetic evidence for the binding order remains to be established”. We have amended the caption to reference work on MelB and the DASS family, as follows:

“We hypothesise that Na⁺ binds first **(similar to other Na⁺-solute symporters^{42,43})** and this primes the transporter for subsequent carnitine binding, which involves the critical R471 within the C-terminal bundle (orange).”

- On lines 289-290 and 297-298, the locations of pathogenic mutations are noted, but an explicit hypothesis for their mechanism is not provided. These should be commented upon if at all possible, even if tentatively.

The variants described all result in changes from a charged to an uncharged residue, where previously the residue was a part of an ionic network. As our structures show that these networks contribute to gating interactions, we suggest the mechanism of pathogenicity involves disruption to the gating of the transporter. We have added this with the caveat that some of these variants may affect folding, stability or localisation, and noted which of these variants have been found to localise normally.

“Together these interactions link TMs 2,3 and 4 (**Fig. 3b**). R169Q, R169P and R169W are all variants associated with SPCD¹¹, **all of which we hypothesise affect the positioning of both D165 and E220, and subsequently the gating interaction between R459 and E220. It is possible that these variants have impaired folding and localisation, although R169W has previously been shown to localise normally to the membrane⁵⁵.**”

“R282Q is a variant associated with SPCD, **and localises normally to the plasma membrane⁵⁵. Altogether, these data support a pathogenicity mechanism where variants can disrupt transporter gating.**”

- *An extended data figure should be provided to show the density of the side chains highlighted in Fig. 3b-d. This is needed to justify or give confidence in the modeling of these side chains on the periphery of the protein, where the map quality is likely lower.*

We have added this to **Extended Data Figure 5**, panel b.

- *The text (lines 341-342) describes TM11 bending, while figure 3f appears to show TM11 breaks between R471 and G473. Can this figure be modified, or an overlay provided in an extended data figure, to better illustrate this movement?*

We have added an overlay and changed the arrows in **Fig 3f** to better illustrate and make the bending motion more clear. The helix bends about residues R471-L472 in the lower part of the helix and residues P478-Y479 in the upper part (as confirmed by DynDom analysis). The $i + 4 \rightarrow i$ hydrogen bonding pattern remains intact, supporting that these are bends and not breaks.

- *It is difficult to visualize the similarity or differences in the coordination of carnitine and ipratropium. Can the authors provide an overlay of the two substrate-bound structures to better illustrate this point?*

We have added **Extended Data Figure 10** showing views of the binding sites and highlighting structural changes (**b**). We also show in (**a**) the hypothetical positioning of ipratropium within the occluded cavity based on an overlay of the two structures.

- I am confused by the description of ipratropium (page 17), particularly as it relates to possible binding in an occluded conformation. Based on the described structures, it does not trigger the same conformational change as a bona fide substrate. Correspondingly, it does not generate transport currents and blocks carnitine transport currents. Therefore, while the authors do not directly assert it, ipratropium appears to meet all criteria for an inhibitor and the results should be analyzed accordingly.

We thank the reviewer for these observations, and agree this needs clarification. In our experimental conditions ipratropium is acting as an inhibitor, though we cannot rule out very low levels of transport. As in our response to Reviewer 2 below, our data support inhibition, and it is most likely competitive based on our structural data. We have amended the final paragraph (Page 17) to more strongly assert ipratropium is functioning as an inhibitor in our experiments. Lines 402-409 now read:

“These structural insights are mirrored in our functional data. Ipratropium did not elicit any current when applied to oocytes expressing hOCTN2, **and rather acts as an inhibitor of carnitine transport**, with an IC_{50} in the μM range ($615 \pm 174.5 \mu\text{M}$, $n=5$) (Fig. 4). Given that ipratropium bears a positive charge, even uncoupled transport (i.e. without sodium) should result in the generation of current. **While these data do not rule out slow or electroneutral transport, they strongly support that ipratropium primarily binds and blocks the transporter, rather than being efficiently translocated. This is consistent with a mechanism where the R471-carboxylate interaction is required to couple substrate binding to transport.**”

We have further clarified the point of comparing the ipratropium and carnitine-bound states, which is that we think that the transporter can sterically accommodate ipratropium, and that the missing R471 interaction is a more likely driver of inefficient transport. Lines 390-395 now read:

“It is notable that in the cryo-EM conditions here, ipratropium did not induce occlusion like with the native substrate carnitine. A structural overlay of the ipratropium-bound structure and the occluded structure demonstrates that ipratropium **in principle could fit** within the occluded substrate-binding site cavity (Extended Data Fig. 10), **with only minor adjustments to the substrate-binding site to alleviate** a steric clash between the tropane moiety and F443 of TM10. **We propose that the absence of the R471 interaction with ipratropium likely accounts for the lack of occlusion.**”

- The discussion's comments on R471 are confusing and appear contradictory. How can the data from (poor substrate/inhibitor) ipratropium suggest that engaging this residue is critical for transport, while substrate TEA's inability to interact with this residue also highlights its importance?

We agree this is confusing, and have clarified the discussion accordingly. We emphasise that TEA is also a poor substrate (which others have shown to be 0.2%-3% the rate of carnitine transport), and that the inefficient transport of TEA is likely reflected by the lack of R471 engagement. We have amended the discussion as follows. Lines 452-457:

We suggest that this is because ipratropium is able to engage the OCTN2 aromatic cage, but is unable to form a salt bridge with R471. Indeed, OCTN2 has been shown to transport TEA in a Na⁺-independent manner¹⁵, albeit much less efficiently than carnitine (**~0.2-3% transport rate relative to carnitine**)^{18,23}. **Similar to ipratropium**, cationic TEA would also not be able to form a salt bridge with R471, further **supporting** that **the R471 salt bridge is required** for **efficient** transport by OCTN2.

- *Was heterogenous refinement used during data processing? As described in the methods, the authors used an unusual procedure by going straight from ab initio to non-uniform refinement.*

We thank the reviewer for noticing this—heterogeneous refinement was indeed used once for the occluded conformation and once for the ipratropium data processing, so we have amended the workflow and methods accordingly. This was initially excluded as the majority of particle sorting was achieved by multiple rounds of multi-class ab initio reconstruction.

- *Were any other non-junk classes observed during data processing? The maps from ab initio (or heterogeneous refinement) should be included in ED Fig. 1 to give a sense of whether any alternative conformations or species are present in the sample. (Though I appreciate the compact and neat layout of the data processing figure.)*

We have expanded **Extended Data Fig. 1** to include the maps from *ab initio* and heterogeneous refinement. We did observe some non-junk classes where there was density visible for the extracellular domain but not the transmembrane domain (as seen throughout the workflow). However, further classification and refinement of these maps did not yield anything interpretable. Additionally, we note that extensive 3d classification of the largest particle stacks (after 2d classification) did not uncover alternative conformations or species.

Reviewer #2 (Remarks to the Author):

This report details the structure of a carnitine transporter, OCTN2. The cryo-EM structure was obtained at good resolution in an occluded conformation and shows the substrate binding site, as well as a bound Na⁺ ion. In addition, the structure of an inward-facing, substrate-free state was obtained. Finally, the authors demonstrate the structure of an ipratropium-bound inward-facing conformation. Ipratropium is characterized as an inhibitor of OCTN2, since no transport activity was detected. This is very nice work that has several novel aspects. 1) While the structure of OCTs is known (not surprisingly showing the MFS fold), the structure of a member of the OCTN subclade has not been determined yet. While the fold is as expected, this new structure informs on transport mechanism. For example, Na⁺ is bound in the inward-facing, substrate-free conformation, suggesting sequential Na⁺/substrate binding. 2) The Na⁺ binding site is unusual in the MFS family and is connected to a water-filled cavity. The implications of this finding for the mechanism await further study. 3) The inhibitor-bound structure is novel and is interesting, due to its high similarity to the inward-facing structure. Overall, this is sound work that advances the field. I have some minor comments:

-Is Ipratropium a competitive inhibitor? If yes, has this been experimentally shown?

We suggest that it is very likely ipratropium is a competitive inhibitor, as alongside the dose-dependent inhibition of carnitine-activated currents, we see in the structures that ipratropium binds the carnitine site within the transporter. We are aware that our kinetics do not directly show competitive binding, so we have not explicitly labelled it as such.

-Is substrate uptake inhibited by ipratropium? Currents don't measure flow, so they are only an indirect metric of transport.

Although currents do not directly report substrate flux, they do report on transport-coupled charge movement. This method was chosen as we are unable to readily source radiolabelled ipratropium. We can confidently say that in our experiments, uptake of L-carnitine (**Fig 4c**) was inhibited by ipratropium.

-It would be important to show original OCTN2 current traces in the extended data.

We have added current traces for OCTN2 WT, mutant transporters and uninjected controls as **Extended Data Figure 9** as suggested. We note that all raw data for the electrophysiology experiments are also available in the Source Data file.

We thank the reviewers for their helpful, constructive comments, which have led to substantial improvements to the manuscript. We sincerely appreciate the time and expertise devoted to reviewing our manuscript.